# Multimodal Assessment of Cognitive Workload Using Neural, Subjective and Behavioural Measures in Smart Factory Settings

**DOI:** 10.3390/s23218926

**Published:** 2023-11-02

**Authors:** Zohreh Zakeri, Arshia Arif, Ahmet Omurtag, Philip Breedon, Azfar Khalid

**Affiliations:** Department of Engineering, School of Science and Technology, Nottingham Trent University, Clifton, Nottingham NG11 8NS, UK; arshia.arif2021@my.ntu.ac.uk (A.A.); ahmet.omurtag@ntu.ac.uk (A.O.); philip.breedon@ntu.ac.uk (P.B.); azfar.khalid@ntu.ac.uk (A.K.)

**Keywords:** cognitive stress analysis, human robot collaboration (HRC), neuroimaging, EEG, fNIRS, machine learning

## Abstract

Collaborative robots (cobots) have largely replaced conventional industrial robots in today’s workplaces, particularly in manufacturing setups, due to their improved performance and intelligent design. In the framework of Industry 5.0, humans are working alongside cobots to accomplish the required level of automation. However, human–robot interaction has brought up concerns regarding human factors (HF) and ergonomics. A human worker may experience cognitive stress as a result of cobots’ irresponsive nature in unpredictably occurring situations, which adversely affects productivity. Therefore, there is a necessity to measure stress to enhance a human worker’s performance in a human–robot collaborative environment. In this study, factory workers’ mental workload was assessed using physiological, behavioural, and subjective measures. Electroencephalography (EEG) and functional near-infrared spectroscopy (fNIRS) signals were collected to acquire brain signals and track hemodynamic activity, respectively. The effect of task complexity, cobot movement speed, and cobot payload capacity on the mental stress of a human worker were observed for a task designed in the context of a smart factory. Task complexity and cobot speed proved to be more impactful. As physiological measures are unbiased and more authentic means to estimate stress, eventually they may replace the other conventional measures if they prove to correlate with the results of traditional ones. Here, regression and artificial neural networks (ANN) were utilised to determine the correlation between physiological data and subjective and behavioural measures. Regression performed better for most of the targets and the best correlation (rsq-adj = 0.654146) was achieved for predicting missed beeps, a behavioural measure, using a combination of multiple EEG and fNIRS predictors. The k-nearest neighbours (KNN) algorithm was used to evaluate the accuracy of correlation between traditional measures and physiological variables, with the highest accuracy of 77.8% achieved for missed beeps as the target. Results show that physiological measures can be more insightful and have the tendency to replace other biased parameters.

## 1. Introduction

Traditional industrial robots are being used in manufacturing sectors for physically demanding tasks requiring high precision or numerous iterations [1]. However, conventional robots are not well-suited when the customisation of products involves variable levels of automation [2]. A completely automated environment is highly desirable but challenging to achieve because of constraints like cost and resources, whereas entirely manual systems, even though adaptable, can be highly erroneous and time-consuming due to a human worker’s exhaustion and stress [2]. To achieve a customised level of automation by leveraging the speed, precision, and power of robots, the focus of research is now converging towards human–robot collaboration (HRC) in the context of industry 5.0.

Collaborative robots, also known as cobots, are specifically designed to operate with humans in the same workspace. In HRC, one of the most significant benefits is that cobots can ensure human worker safety by controlling their motion as they sense the position of a human operator around them [3]. Major applications of cobots include machine tending, assembly tasks, pick-and-place jobs, palletising, etc. [4].

Even though the introduction of cobots has upgraded performance, HRC has led to several issues related to HF and ergonomics in terms of mental stress and cognitive workload. A human worker can feel disconcerted and nervous by a robot’s size and unexpected mobility [5]. Numerous factors, including human worker resistance to cobots, prolonged task durations, fatigue, and safety concerns posed by the motion and speed of cobots’ physical components, can contribute to these challenges [5,6]. This overwhelming experience for a human worker can lead to unfavourable productivity, as an HRC task is prone to more human errors and mishaps in a panic condition [6]. Under these circumstances, there is a need to develop human-centred industrial setups that prioritise HF considerations to reduce human workers’ anxiety level, hence enhancing system efficiency [7]. Therefore, monitoring the cognitive stress of a human operator accurately is an essential goal to achieve.

Researchers have been estimating mental stress factors using behavioural, subjective (i.e., questionnaires), and objective measures (i.e., physiological data) [8]. To investigate the impact of cobot speed and path predictability on a human worker, two concurrent tasks, manufacturing components and a quality control task in a 3D virtual environment, were carried out by Koppenborg et al. (2017) [9]. The NASA task load index (NASA-TLX) and state-trait anxiety Inventory (STAI-S) questionnaires have both been used as subjective measures to assess the mental workload and anxiety of human workers, respectively. Alongside, as a physiological measure, mean inter-beat intervals were analysed using a chest sensor [9]. The impact of graphical signage on a human worker’s performance and anxiety was evaluated for a bolt extraction task from tubes using the KUKA robot by Eimontaite et al. (2019) [1]. The negative attitude towards robot scale (NARS) and robot anxiety scale (RAS) [10] have been used to quantify human workers’ cognitive strain, whereas facial expressions have also been monitored, using cameras, as behavioural stress indicators [1]. Another 10-item questionnaire, the perceived stress scale (PSS-10), was employed by Björling et al. (2019) to evaluate the perceived stress of a teen during robot interaction [11]. Rossato et al. (2021) assessed the subjective experience of senior and younger workers in HRC using the technology acceptance model (TAM) and NASA-TLX for measuring a user’s acceptance of technology and cognitive task load, respectively [12]. NARS was also administered by Gervasi et al. (2022) to monitor the mental workload of an operator while performing a collaborative assembly task [13].

A relatively simple secondary task is often introduced alongside a major primary task to increase the complexity of the experiment, so that the decision-making aspect, in the context of reaction time, of a human worker can be assessed as a behavioural measure [8]. Initially, Zakeri et al. (2021) designed an experiment to replicate an industrial situation for studying HF pertaining to a human worker’s mental stress in a factory environment. A primary cobot–Stroop task accompanied by a secondary pedalling activity was designed considering performance parameters, i.e., different speeds of cobot motion, cobot payload capacities, and task complexities [8]. Subjective, behavioural, and physiological measures were taken into consideration while devising this scheme. Since the stress assessment questionnaires are filled in by the human worker at the end of the task, there are no means to objectively observe the changes in stress levels during the task via subjective measures. Therefore, EEG and fNIRS signals, as physiological measures, along with the NASA-TLX questionnaire, were utilised by Zakeri et al. (2022) to estimate the cognitive stress of a factory employee in real time throughout the task execution, in terms of experiment performance parameters [14]. Results of this study indicate that task complexity is directly related to heart rate, beta and gamma average band power, and left prefrontal cortex activation, whereas it is indirectly related to alpha band power.

In factories, especially in manufacturing environments, there is a need to consider neuroergonomics and HF to create a stress-free atmosphere for employees working with cobots. When a human worker is interacting with a cobot, there will inevitably be moments of hesitation, fear, and anxiety. Developing a scheme to determine cognitive stress and implementing required measures to alleviate a human worker’s mental load can lead to a healthy change for factory employees, thus boosting their confidence, motivation, and determination. Such a strategy can ensure emotional, physical, and mental safety around cobots, hence reducing human workers’ insecurity toward robots. The first goal of this study is to analyse how task properties (e.g., speed, complexity, etc.) affect the perceptual state of factory workers under cognitive load conditions by monitoring this impact in subjective, behavioural, and physiological measures. Therefore, this research intends to examine the effect of varying levels of task properties, by considering them individually, in pairwise combinations, and collectively, on the mental stress of factory employees.

Physiological measures can be more informative, as they can provide real-time information while the experiment is being performed [15], whereas other conventional metrics have multiple factors due to which their results can be biased. The gap lies in the fact that physiological measures have not been established yet. Therefore, the second aim of this study is to find out whether the physiological measures can predict the traditional ones, i.e., behavioural and subjective measures. The goal is to understand how well neuroimaging is able to predict traditional measures.

## 2. Methodology

The experiment for this study was formulated, using cobots, in such a way that it mimics the situation of a factory worker in an automated industrial setup. Brain signals of the human operator were acquired to get neurovascular information during the task implementation.

A brain–computer interface (BCI) was implemented to obtain brain data, such as EEG and fNIRS signals, during the experiment. These techniques can aid in the acquisition of attributes which are useful in the assessment of cognitive stress and anxiety [16]. Both BCIs are categorised as non-invasive neuroimaging methods. fNIRS measures the changes in local cerebral haemoglobin concentration by sensing the light absorption in the cortex, using an infrared light-emitting source and detector combination [17]. Heart rate, which is expected to vary with the occurrence of stress [18], can be extracted from fNIRS data [19]. EEG is an electrophysiological measure that analyses the neural activity of the brain using electrodes located at the head surface [20]. These variables can come in handy for monitoring the vigilance and attention of a person [16] while performing the required experiment.

### 2.1. Experimental Paradigm

For this research, a pick-and-place task is selected which involves the decision-making of a human worker. An environment is created where the human operator needs to collaborate with a cobot and adjust his performance speed to match that of the cobot. The complete experiment is composed of two tasks to be performed concurrently. These factors can collectively induce cognitive stress in a person.

Participants were initially selected based on the criteria that only healthy adults, male and female, within the age range of 18 to 55 years, without any neurological disorder, head trauma or other head injury background, could take part in the experiment. Age was used as a criterion since a child’s and an older person’s brain characteristics are different to those of a normal adult brain. Any motor disability (upper or lower extremity injury or disability) was also used as an exclusion criterion, as the experiments involved testing participants performing bimanual motor activity. Data were acquired from a total of 13 selected participants, from among university students and staff from PepsiCo International Limited, but only data from 9 participants were used for this research. The remaining data were discarded due to their poor quality. The technological background of participants was such that 84.6% of them were social media users, 100% were smartphone users and 30.7% were already familiar with robot interaction. The total task duration was approximately an hour for each participant. The Stroop task, which is a standard task for assessing a person’s control of his cognitive behaviour [21], was adapted in this study. Forty cubic boxes were provided for this experiment, each with an equation and a colour name (red, blue, or green) printed on it in a different coloured ink. For instance, “blue” might be written on a box with red-coloured ink. In this research, the Stroop task, in collaboration with a robot as shown in Figure 1, was considered a primary task. The participant has to categorise the boxes based on the rule that if the equation is correct, he has to consider the colour of the text rather than the word written on it; otherwise, he has to focus on the name of the colour. A cobot is involved in this task in such a manner that it has to pick a cubic box from a corner, where all boxes are present initially, and pass it on to the human participant. The participant has to take the box and place it in its designated place on the workstation, according to the rule described above [14]. The participant must match his speed with that of the cobot; otherwise, he might be occupied in placing the previous box, whereas the cobot would not wait for the human and drop the next box. This can be considered an error on the human’s part, and consequently, performance can be reduced as a result of slow decision-making. Each participant has to carry out the task in eight different experimental conditions, termed episodes, determined by different combinations of the cobot payload capacities, task complexities, and cobot movement speed.

To imitate an industrial scenario, where the human worker has to make decisions while performing multiple tasks concurrently, the cobot–Stroop task was coupled with another comparatively simpler task. A secondary task was introduced with the primary cobot–Stroop task to vary the task complexity from low to high, in different episodes of the task. The episodes in which complexity needs to be higher must contain the presence of this accompanying task. As a secondary task, beeps are played after 500 to 1000 ms intervals, and the participant has to respond to these beeps by pressing a foot pedal, for each beep. Human error can be calculated by counting the number of missed beeps. Participants’ response rate to beeps and reaction time are considered behavioural measures.

The flow of the experiment is such that the participant’s brain data are collected throughout the 60 min of the experiment. After complete setup at the beginning of the experiment, only the secondary foot-pedalling task is performed for 2 min, as a baseline case. Subsequently, episode 1 of the cobot–Stroop task, where the cobot speed, payload capacity, and task complexity are all at a low level, is performed for 4 min. Afterwards, a rest episode is conducted, where the participant must sit in a relaxed condition for 2 min, followed by filling out the NASA-TLX form. NASA-TLX is used as a subjective measure and the participant is directed to fill out the form after each episode of the experiment. Then, the flow of the experiment is followed as shown in Figure 2. In each episode, parameter levels are set as either low or high, as illustrated in Table 1. The cobot speed for high and low levels was 1 m/s and 0.6 m/s, respectively. Universal robots of two payload capacities, 3 kg (low) and 5 kg (high), were employed in the experiment. The episodes with high task complexity included the secondary task along with the primary task whereas the ones with low task complexity included only the primary task.

### 2.2. Data Acquisition

For this research, EEG and fNIRS signals were collected simultaneously to obtain the brain’s electrical signals and hemodynamic activity, respectively. EEG signals were acquired using TMSi Mobita wireless data acquisition, at a sampling frequency of 2000 Hz [22]. Data were recorded from 19 EEG electrodes, positioned on the scalp according to the international 10–20 system. fNIRS data were recorded using Artinis Octamon, at a sampling rate of 10 Hz, using 8 channels [23]. The distance between the transmitter–receiver pairs is 20–30 mm. These selected channels cover the left frontal area between FP1-F3-F7 and a similar frontal region on the right side too.

During the experiment, data for 3 behavioural measures were collected, i.e., reaction time, missed beeps, and the Stroop task error rate. The Stroop task error rate was not used in this study as it did not significantly distinguish between different episodes. Only reaction time and missed beeps were used as behavioural metrics. During high-complexity episodes involving the secondary tasks, reaction time was recorded for each beep when the participant pressed the pedal. These values were recorded for each episode. Simultaneously, if the participant missed a beep, it was counted. At the end of each episode, the number of missed beeps was counted for each participant. The significance of the reaction time is that it can be an indicator of stress, as it is measured in the high-complexity episodes involving the secondary task too and can show the delay in the response time of a participant if he/she is mentally stressed. Missed beeps are also measured in the high-complexity episodes. When the participant is preoccupied mentally with the primary task, it would be challenging for him/her to cope with the beeps. Therefore, an increase in the frequency of missed beeps would signify an increase in the cognitive workload of the subject. Additionally, the Stroop task error was also calculated for the complete experiment. A rise in the number of incorrect categorizations would indicate higher levels of mental stress. The Stroop task error could not produce informative results, so it was excluded from this study.

NASA-TLX is a multidimensional standard scale that can be used to measure stress, fatigue, and consciousness of a person, etc. [24]. The final score of cognitive workload depends on the weighted average of 6 factors, rated by the participant at the end of the experiment. These factors include mental demand, physical demand, temporal demand, performance, effort, and frustration level [25]. Marking from 0 to 100, having a step increment of 5, can be given by the participant for each factor [26]. NASA-TLX was used in this research as a subjective measure where each participant/subject has to score each factor on the form, after the completion of each episode. Then, the average of all 6 factors is considered as the overall score associated with each subject.

### 2.3. Data Pre-Processing and Artefacts Removal

Both EEG and fNIRS data were pre-processed to get clean signals for extracting the required variables. The ICA-based method was used to remove non-brain signals and artefacts from the raw EEG recordings [27]. A zero-phase Hamming windowed-sinc FIR filter, a band-pass filter of 0.16–40 Hz, was also applied to the data for reducing the high-frequency artefacts and the impact of EEG drift. Moreover, EEG signals from all channels were down-sampled from 2000 Hz to 200 Hz. EEG has multiple frequency bands with different ranges, i.e., delta (0.5–4 Hz), theta (4–8 Hz), alpha (8–12 Hz), beta (12–28 Hz), and gamma (28–50 Hz) [28]. The expected behavioural states, i.e., deep sleep, deep meditation, awake but relaxed, cognitive thinking, and unifying consciousness, are associated with delta, theta, alpha, beta, and lower gamma bands, respectively [29]. Frequency band power (FBP) was analysed for these 5 bands, as shown in Figure 3.

Likewise, in fNIRS data, there can be multiple technical and biological artefacts that can distort the data [30]. One of the technical artefacts can be different calibration and coupling of optodes, causing differences in the average amplitudes of channels, which may stay constant during the task. Biological artefacts include head movements, which cause sudden distortions due to the affected coupling of optodes [14]. Then, there is muscle oxygenation, occurring particularly close to the temporalis muscle, which results in a long-lasting peak of high amplitude. Transient significant deflections, from variations in blood perfusions, are a consequence of the participant’s upper body movement. Blood flow in superficial (non-cerebral) tissue, Mayer-waves artefacts at the frequency of around 0.1 Hz, and systemic heartbeat at approximately 1 Hz, are all categorised as biological artefacts [14].

To acquire artefact-free and accurate data, initially, each channel’s signal was passed through a band-pass filter ranging from 0.15 Hz to 0.5 Hz, which minimised heartbeat activity and some slow components. Then, the Beer-Lambert law was applied to filtered recordings to get them in the form of oxy-haemoglobin (HbO) and deoxy-haemoglobin (HbR) concentrations [8]. Since there is an overlapping time scale of cerebral activity and Mayer waves, band-pass filtering could not be used to minimise the latter. However, Mayer waves are not task-driven, and the waves of different participants are not synchronised; thus, their effects were not expected to influence our study, and they tended to cancel out in group averages [31,32]. To remove high-peaked artefacts caused by movement or muscles, outliers in all channels’ recordings were identified and eliminated if they were more than three scaled median absolute deviations (MAD) far from the median [14]. To reduce the amplitude differences between channels which might be due to a particular subject or optical coupling, haemoglobin concentration signals were normalised as a result of dividing them by the standard deviation (SD) of the preceding rest interval. For episode 1, signals were normalised by using initial rest duration. To diminish systematic components, which can appear in the whole signal, the complete recording for each channel was divided by its average [14]. Following these steps can help mitigate the above-listed types of artefacts, making the data more accurate and indicative of only cerebral activity. The complete process of artefact removal and feature extraction for EEG and fNIRS data is depicted in Figure 3.

Figure 4 and Figure 5 show the comparison of raw and processed data for EEG and fNIRS, respectively. Figure 4a shows the raw EEG data for all 19 channels, whereas artefact-free EEG data are visible in Figure 4b. Figure 5a displays the raw data for fNIRS, whereas in Figure 5b the concentration changes in oxygenated haemoglobin for channels 1–8, of subject 8, episode 2, are shown with y-offset for visibility. The signal-time segment is automatically marked as an artefact and excluded from further analysis, as shown by grey shading.

### 2.4. Use of Machine Learning and Statistical Analysis for Prediction of Traditional Measures Using Physiological Measures

Artificial intelligence (AI) has an area called machine learning that focuses on constructing algorithms and models that allow computers to learn from the surrounding environments and make predictions or decisions without the need to be explicitly programmed [33]. To predict traditional measures using neural measures, two machine learning techniques were employed: linear regression [34] and artificial neural networks (ANN) [35], as shown in Figure 6. For both algorithms, the brain data, i.e., EEG and fNIRS features, were used as predictors, whereas subjective measure (NASA-TLX questionnaire) and behavioural measures (missed beeps and reaction time) were considered as targets. For each target, multiple combinations of predictors, i.e., EEG features alone, fNIRS features alone, and EEG and fNIRS features collectively, were considered.

Statistical analysis was carried out to assess the performance of both models. The Signrank test, used to calculate the *p*-values, is a non-parametric method suitable for analysing non-normally distributed data to evaluate the significance of differences between paired samples. Subsequently, the Bonferroni correction (threshold = 0.01) was applied to address the multiple comparison issues and uphold the integrity of the statistical inferences.

A variation of the R-squared metric called adjusted R-squared is used to analyse how well the regression model fits the data. When a big set of predictors is introduced or there are numerous variables, an unadjusted R-squared might be deceptive while evaluating the model. An increase in the squared value due to more predictors can result in a poorly fitted model being presented as a well-fitted one. This misleading behaviour of R-squared can be catered to using adjusted R-squared while keeping the number of independent variables in the account [36]. Adjusted R-squared was used as a metric to compare both models in the context of fitting the data, as it restricts the addition of extraneous features. The value of the adjusted R-squared only rises when a model’s performance improves due to the addition of a new predictor [37]. Adjusted R-squared can be calculated using the following expression (1):(1)$$R_{adj}^{2}=1-(1-R^{2})[\frac{n-1}{n-m-1}]\;$$
where *m* is the number of independent variables and *n* denotes the sample size of data [36].

In this study, KNN was also used, as shown in Figure 7, to assess the accuracy of correlation between EEG-fNIRS data and conventional measures, i.e., subjective and behavioural measures.

## 3. Results and Discussion

The effect of task properties like robot payload capacity, speed, and task complexity were investigated in this study and were evaluated using behavioural, subjective, and physiological measures. The effects of individual, pairwise combinations and a combination of all parameters were examined for all stress assessment measures. Results for each category are discussed/presented below. Furthermore, machine-learning techniques were used to find the correlation between the traditional measures (subjective and behavioural measures) and neural measures (EEG and fNIRS variables).

### 3.1. Impact of Performance Variables on Cognitive Stress Considering Subjective Measure—NASA-TLX

Figure 8 shows the impact of all performance variables on NASA-TLX scores. The box plot was constructed by using the NASA-TLX score of all participants for all episodes. Significant differences can be seen between multiple episodes of the task in Figure 3, indicated by asterisks (*). Episode 2 has a higher NASA-TLX score as compared to episodes 4 and 7. Payload capacity and speed are low and high, respectively, in episodes 2 and 4 but the only factor that has changed is task complexity, high for episode 2 and low for episode 4. In episode 7, however, even though payload capacity is high, the effect of low speed and low complexity on cognitive stress is visible in terms of a low NASA-TLX score, depicting a decrease in mental stress. The difference between episode 2 and episodes 4 and 7 is significant, demonstrating the pronounced effect of complexity on the cognitive state of human workers. The significant difference between episodes 3 and 4 can also be observed where payload capacity has not changed, but cobot speed and task complexity are different for both episodes. Cobot speed is low in episode 3 and high in episode 4, whereas task complexity is high in episode 3 and low in episode 4. The higher NASA-TLX score in episode 3 highlights the increase in cognitive stress of human workers because of high task complexity. Furthermore, episode 5 also has significant differences with episodes 1, 4, and 7. Here too, the impact of high task complexity is more visible than other parameters in the form of a high NASA-TLX score for episode 5. Similarly, episode 8 exhibits a significant difference with episode 4. Task complexity is the only dissimilar parameter in episodes 8 and 4 that has raised the cognitive load of a human worker.

The high task complexity episodes, episodes 2, 3, 5, and 8, exhibit comparatively high scores. This is evident from Figure 8 in the form of significant differences between high-complexity episodes and low-complexity episodes. This finding is also backed up by Capa et al. (2008), who researched the assessment of the impact of task complexity on subjective, physiological, and behavioural measures [38]. This pattern exhibited by NASA-TLX scores reflects the stressed mental state of human workers in high-complexity episodes. The NASA-TLX score for the 8th episode is not very high as compared to the other high-complexity episodes, as the effect of learning can be seen here. Till the last episode, the participant becomes familiar with the task and interaction with the cobot. In conclusion, it is evident from NASA-TLX scores and significant differences in these episodes that task complexity has the most impact on the mental state of human workers, followed by speed. Change in payload capacity did not prove to have made any notable difference in the mental stress of participants.

### 3.2. Impact of Performance Variables on Cognitive Stress Considering Behavioural Measures—Reaction Time and Missed Beeps

In Figure 9 and Figure 10, the pedal-only and four high-complexity episodes are illustrated because only these episodes involve the secondary foot-pedalling task. In the high-complexity task episodes, especially when cobot speed is also set to a high value, participants can miss a comparatively larger number of beeps because of the “inattentional deafness” phenomenon [39]. Due to the high attentional demands of the primary task, a participant might fail to hear the beeps which are otherwise audible. In Figure 9, episode 2 has the highest number of missed beeps detected, which is a consequence of cognitive load induced by collaborating with cobots during high task complexity and at high cobot speed. The numbers of missed beeps for episodes 3 and 5 are comparable with the pedal-only episode, as the cobot speed is low in both these episodes. Even though all parameters are set to a high value in episode 8, most of the participants missed a smaller number of beeps. This can be due to the effect of learning on their cognitive state or the subject’s disengagement from the primary cobot–Stroop task. The subject’s attention can be biased more toward the beeps when all variables are set to a high value. Consequently, a smaller number of missed beeps can be observed in episode 8. No significant difference between any episodes can be seen in the case of missed beeps. It is clearly visible from Figure 3 that the number of missed beeps has increased but not significantly when comparing pedal-only and the other high-complexity task episodes.

In Figure 10, reaction time can be seen to be high for episode 2 as a result of anxiety caused to human workers by high cobot speed and task complexity. A slight reduction in mental stress of participants can be observed when the speed of the cobot decreased in episode 3. This is evident from the decrease in reaction time for episode 3. In episode 5, reaction time has the highest value. This can be a reflection of the tiredness and fatigue of the subjects as the experiment proceeds over time. Here too, episode 8 shows the effect of learning, as the reaction time for pedalling has reduced, even though all parameters have high values. The differences in pedal-only tasks in episodes 2 and 3 are significant in the case of reaction time.

### 3.3. Impact of Performance Variables on Cognitive Stress Considering Physiological Measures—EEG and fNIRS

Frequency band power information was extracted from EEG data to analyse the effect of payload capacity, task complexity, and cobot speed on stress level/cognitive load during task performance. Initially, the average relative power for all episodes in which the payload capacities change from either low to high or vice versa was compared. For example, episode 1 and episode 7 having low and high payload capacity, respectively, were compared. Similarly, episodes 2 and 8 were placed side by side in Table 2 to see the difference. In these episodes, the other two performance measures, i.e., cobot speed and task complexity, are constant, either low or high. Likewise, a comparison of average relative power in multiple frequency bands for different episodes was performed, based on each performance measure. Statistical analysis helped to identify the significant differences, indicated in Table 2, Table 3 and Table 4 by *p*-values highlighted in red ink. In the comparison of episodes 1 and 4, where speed is changing but task complexity is low, a significant difference can be observed in the gamma band (Table 2). Similarly, in episodes 5 and 8, just the cobot speed is changing, but task complexity is high. Consequently, a slightly notable difference in alpha and initial beta bands can be observed, but it is not significant. For episode 1 against episode 3, only task complexity is changing, but for low speed and low payload capacity, significant differences are visible in theta, alpha, and beta 1 bands. Likewise, the comparison of episodes 6 and 8, where task complexity is changing but payload capacity and cobot speed are set to high values, shows significant differences in alpha and beta 1 bands. From the results shown in Table 2, it can be observed that mostly alpha and initial beta bands are affected by induced mental stress in the participants. It can be clearly seen from Table 2 that the impact of complexity is the greatest, especially in the alpha and beta bands. Speed also showed some impact in the gamma band, but no significant difference can be observed in the case of payload capacities.

Afterwards, average relative power is analysed for the pairwise combination of these performance parameters in Table 3. In Table 3, episodes 1 and 6 are first compared where task complexity is set to a low value for both, whereas payload capacity and speed are changing. A significant difference in the theta band can be observed, demonstrating the increase in cognitive stress. Similarly, episodes 3 and 8, with only high task complexity being constant, were compared, but there was no notable effect observed in this case. Now, all episodes with varying payload capacity and task complexity but the same speed level were observed. For episodes 1 and 5, where speed is set to low, a significant difference in the theta band is visible, but when episodes 4 and 8, with a high-speed setting, are compared, no significant difference can be seen in the theta band; rather, it is present in the alpha and beta1 band. This shows the combined impact of payload capacity and task complexity on the mental stress of the participants. Then, episodes 1 and 2 are compared to see the effect of changing speed along with task complexity. Significant differences can be seen in the theta, alpha, and beta1 bands. Likewise, in episodes 7 and 8, where payload capacity is high with other variables changing, the difference is significant in almost all the bands except delta and beta 4. This discussion from Table 3 concludes that simultaneous changes in cobot speed and task complexity have the most impact on a subject’s mental state. Most cognitive stress is induced by this pairwise combination of cobot speed and task complexity. The impact of speed and complexity combined seems to be comparatively higher than other pairwise combinations.

Finally, the impact of all three measures changing can be seen in Table 4. A significant difference can be seen in the theta, alpha and initial beta bands [40]. This significance is because when a human worker is under cognitive stress, the alpha and initial beta bands are the most affected. The alpha band is associated with a person in a relaxed condition, whereas the beta and gamma bands are linked to cognitive tasks or stressful conditions [41].

Similarly, the impact of task properties (i.e., payload capacity, cobot speed and task complexity) on fNIRS signals can be seen in Figure 11. Box plots were constructed for averaged activations in the right and left prefrontal cortex. Figure 11a has all task episodes on the *x*-axis for both cortices, whereas HbO concentration is on the *y*-axis. In terms of cortices, a relatively higher concentration of HbO can be observed in the left prefrontal cortex, especially in high-complexity episodes (episodes 2, 3, 5, and 8). This argument has also been supported by the findings of Zhu Q et al. (2021) for an industrial valve operation task [42]. Episodes 3 and 8 have the highest values of HbO concentration. Even though episode 3 has low payload capacity and low cobot speed, due to high task complexity, distinguishable mental stress was induced. All variables have high values in episode 8, resulting in comparatively higher HbO values for all participants than in the remaining episodes. Similarly, in Figure 11b, where performance measures (low and high levels) are plotted against HbO concentration, a comparatively better picture is presented by the left cortex. The impact of high complexity and high speed on activation appears to be more pronounced, whereas payload capacities have no noticeable effect. Task complexity seems to have the most impact on the left cortex of participants’ brains. This is congruent with the left hemisphere’s established dominance during attention tasks and general movement [43].

In Figure 11c, HbR concentration is depicted against all episodes, for both cortices. A similar trend can be seen here, in that high-complexity episodes affect HbR concentration more in the left cortex, whereas no specific trend can be seen in the right cortex. Relatively more impact of speed on HbR concentration is illustrated by Figure 11d, but there is no substantial effect of task complexity and payload capacity on HbR concentration.

### 3.4. Prediction of Performance and Subjective Experience from Physiological Variables

If it is possible to predict traditional subjective and behavioural measures using neural data, the former may be supplemented or even replaced by neural parameters. The idea of this study is to investigate how well frequency band powers from EEG signals and pre-frontal oxygenation (activation) from fNIRS signals can predict traditional measures. Multiple combinations of features were considered to apply machine learning techniques. Initially, only EEG features, i.e., frequency band powers, were used, followed by only fNIRS features, i.e., HbO and HbR concentrations, to observe the prediction capability of these predictors alone. Afterwards, combinations of EEG and fNIRS features were used to see if prediction accuracy can be improved or not.

To find out the correlation between neural measures and conventional ones, linear regression and artificial neural networks (ANN) were applied. For these two techniques, the predictors are EEG and fNIRS features (i.e., frequency band powers for EEG bands like delta, theta, alpha, etc., and HbO and HbR concentrations from fNIRS recordings) whereas the targets are subjective measures (NASA-TLX score) and behavioural measures (missed beeps and reaction time). The relatively better and more informative results for multiple combinations of predictors are shown in Table 5. Statistical analysis has provided better results with regression, which are mentioned in the table in the form of adjusted R-squared (rsq-adj) values, a metric used to calculate the correlation. ANN has not performed well in most cases, as our data set is small and ANN requires plenty of data to fit the data [44]. The metric used to assess the performance of ANN is R-squared.

In this research, individual and multiple combinations of EEG and fNIRS variables were employed as features to predict subjective and behavioural measures. Initially, EEG and fNIRS features alone were used as predictors for each target, followed by their collective use. Only high-complexity episodes were used for behavioural targets, i.e., missed beeps and reaction time, because these are only being measured in the presence of secondary foot-pedalling tasks. The highest, lowest, and middle adjusted R-squared values are highlighted in the table using a (~). A combination of multiple EEG frequency bands (theta, alpha, beta1, and beta2) with HbO and HbR concentrations has given the best correlation for predicting missed beeps (rsq-adj = 0.654146). The medium value (0.34464) for correlation is also observed for a combination of EEG and fNIRS for NASA-TLX as a target. These two results show the significance of combining these EEG and fNIRS features, specifically, for the regression model to fit the data. The lowest rsq-adj value achieved was 0.065, using just the beta band to predict reaction time. This smallest value shows the effect of omitting the informative bands, i.e., theta and alpha. Without these frequency band powers, the lowest prediction was achieved for reaction time. Scatter plots for these highest, middle, and lowest values of rsq-adj are illustrated in Figure 12, Figure 13 and Figure 14. When the performance of ANN and linear regression models using R-squared is compared, ANN only performed better in the case of reaction time as the target, with all beta frequency band powers as features (R-squared (linear regression) = 0.215870 and R-squared (ANN) = 0.235478). In Figure 14, the adjusted R-squared is approximately zero for the linear regression model, demonstrating the non-linear correlation between the beta band powers and the reaction time. Therefore, this correlation was missed by the linear regression model, but it was picked up the ANN algorithm.

Adjusted R-squared values for missed beeps are generally higher than the other two targets. Induced cognitive load is most evident from the results of missed beeps and has proved to be a better and more informative target. Reaction time has shown minimum correlation for almost all combinations of EEG and fNIRS. The overall analysis of these results shows that a combination of EEG and fNIRS as features proves to be better for prediction rather than individual EEG or fNIRS features [45].

### 3.5. Classification Results Using KNN for Prediction of Behavioural and Subjective Measures Using EEG and fNIRS Features

A classification technique, KNN, was implemented to model the prediction of conventional measures using physiological measures as a classification problem. It is meaningful to find the predictability using adjusted R-squared (as shown in the previous section), but the accuracy of prediction cannot be calculated. By using KNN as a classifier, it is possible to report the accuracy of predictors. Table 6 describes the results obtained for analysing KNN as the classifier. For this classification too, the impact of individual and multiple combinations of EEG and fNIRS was observed, to predict subjective and behavioural measures. In these results, two types of average accuracies were listed for two cross-validation methods. The first one is where one subject is left out as test data, and in the second case, the same subject is left out from each episode as test data, to calculate the average accuracy. *p*-values were found using statistical analysis to report the statistical significance of the classifier, followed by the application of Bonferroni correction (threshold set to 0.01) and listed in the fourth column of Table 6. All significant cases are highlighted using ** in the table.

It can be observed from Table 6 that for each target, different combinations of EEG and fNIRS were used as predictors. For NASA-TLX as the target, the highest accuracy, 69.4%, was achieved by using all EEG frequency band powers as features while leaving one subject out for testing (first cross-validation method), whereas the second cross-validation method resulted in the highest accuracy of 58.3% when only HbO features are used. Statistical significance of classification using KNN shows that the model acquired by using HbO features alone has the most significance (*p*-value = 0.0016). This emphasises that more accurate results are achieved using the second cross-validation method. Likewise, for reaction time as a target, the first cross-validation method produced the highest accuracy of 65.6% by using all EEG frequency band powers, whereas employing the second method resulted in the highest accuracy of 65.6% by using a combination of all EEG frequency band powers and HbO features. The highest significance can be observed for the model trained using a combination of EEG and fNIRS features (*p*-value = 0.0344). Again, more trusted results can be seen using the second cross-validation method. For missed beeps as the target, classification using all combinations of predictors showed statistical significance, but the *p*-value for most significant cases was acquired for two combinations of predictors, i.e., the combination of all EEG frequency bands and HbR (*p*-value = 0.0012) and HbR features only (*p*-value = 0.0012). Both cross-validation methods showed the highest accuracies for these two combinations of predictors.

From these observations, firstly, it can be deduced that the second cross-validation method (leave the same subject out for each episode for testing) has proved to produce more trusted and true results. Secondly, a noticeable fact from Table 6 is that the combination of EEG and fNIRS features produced significant results, whereas, in the case of NASA-TLX and reaction time as targets, individual features failed to produce significant average accuracies. This aspect reveals that generally multimodal data (EEG-fNIRS) have the ability to yield better average classification accuracy [46]. Thirdly, the missed beeps proved to be a better target, as they produced significant results with all combinations of EEG and fNIRS [39]. These statements are completely aligned with the observations from previous Table 5.

## 4. Challenges and Future Work

During this study, a few challenges were identified. Multiple technical issues with the EEG and fNIRS devices were faced while conducting the experiment. Data for some participants could not be collected because of performance issues of devices and software used for EEG and fNIRS data collection. The EEG-fNISR headcap did not fit properly on some participants’ heads, causing poor connection of the EEG electrodes and fNIRS optodes with the participants’ heads. Some participants had very frizzy and volumized hair, which made it impossible to collect data from their heads. Due to such issues, the data of four participants had to be discarded for this study as they were of poor quality and the dataset reduced from 13 to 9 participants’ data.

The outcomes of this study demonstrate that improved accuracy for mental stress assessment can be achieved using a combination of physiological measures, which can accurately predict the behavioural and subjective measures. This conclusion emphasises the importance of considering additional physiological parameters while estimating cognitive workloads. In the future, a more comprehensive analysis of the mental workload will be possible by incorporating additional physiological, subjective, and behavioural measures like gaze and facial expression monitoring, galvanic skin response (GSR), etc. Larger datasets can be used to acquire more convincing results. Furthermore, other machine-learning techniques can also be used to examine the correlation between physiological and conventional measures, as a comparison with already employed techniques. An aspect of learning was observed in this study, causing a slight increase in performance and a decrease in fatigue and anxiety, as is clearly evident from NASA-TLX scores, missed beeps and reaction time. Further in-depth investigation in this regard can be carried out. This analysis can also be extended to different tasks or populations, i.e., various age groups or occupations. This research can be a step forward to investigate real-time cognitive workloads to provide feedback to people doing stressful jobs.

## 5. Conclusions

In this study, cognitive workload assessment was performed for factory employees in manufacturing setups. To achieve this aim, an experiment was conducted to mimic a multitasking, human-robot collaborative scenario in industries. This experiment was used to estimate the mental stress of a human worker using subjective, behavioural, and physiological measures for different performance parameters, i.e., task complexity, cobot speed, and payload capacity. Initially, the impact of change in individual parameters was assessed for physiological, behavioural, and subjective measures. Then, the effect of change in pairwise combinations of performance parameters was observed followed by the impact of change in all parameters at the same time. Task complexity and cobot speed proved to be more effective factors in the mental stress of a human worker. A correlation between physiological and other conventional measures was also determined using regression and ANN. Regression produced better results showing that missed beeps (a behavioural measure) have the tendency to induce a greater cognitive load. KNN was used to evaluate the accuracy of using multiple combinations of physiological variables as predictors for classification. Two cross-validation techniques were used for this analysis, i.e., leave one subject out for testing and leave the same subject out from each episode for testing. The statistical significance of classification supports the outcomes produced by the second cross-validation method. Our results show that physiological data successfully predict the subjective and behavioural measures to a great extent and missed beeps again proved to be the best target. This research has established the fact that physiological measures are successful in quantifying cognitive stress in HRC environments, as they provide real-time data and unbiased results.

## Figures and Tables

**Figure 1 sensors-23-08926-f001:**
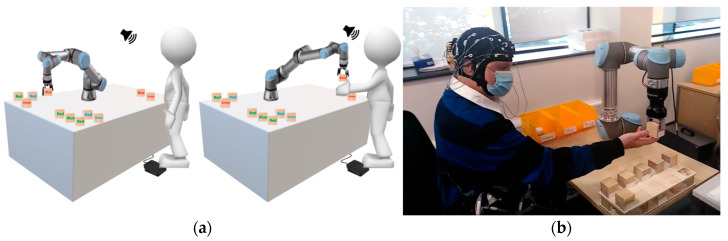
(**a**) Cobot–Stroop Task [8] (**b**) Experimental scene of using cobot while performing cobot–Stroop task [14].

**Figure 2 sensors-23-08926-f002:**
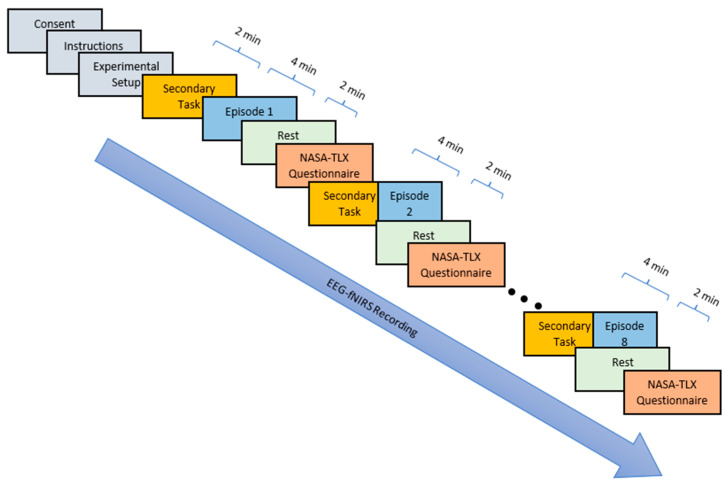
Experimental paradigm for cobot–Stroop task accompanied by secondary foot-pedalling task.

**Figure 3 sensors-23-08926-f003:**
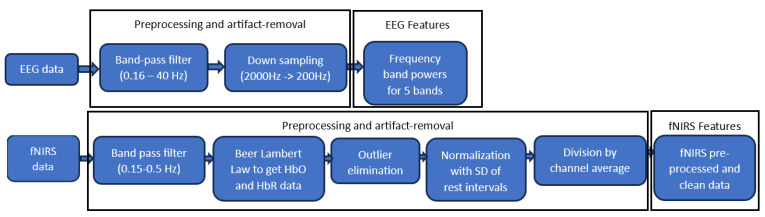
Pre-processing and artefact removal of EEG and fNIRS data.

**Figure 4 sensors-23-08926-f004:**
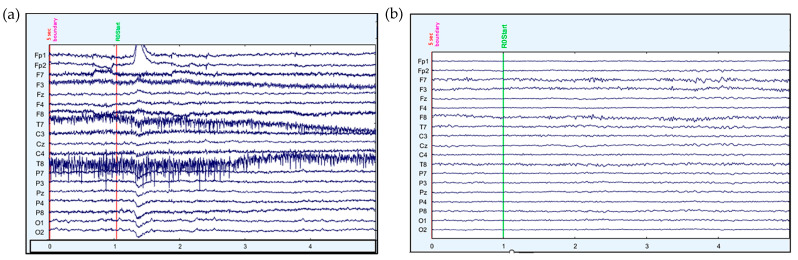
(**a**) Raw EEG data. (**b**) EEG data after artefact removal and pre-processing.

**Figure 5 sensors-23-08926-f005:**
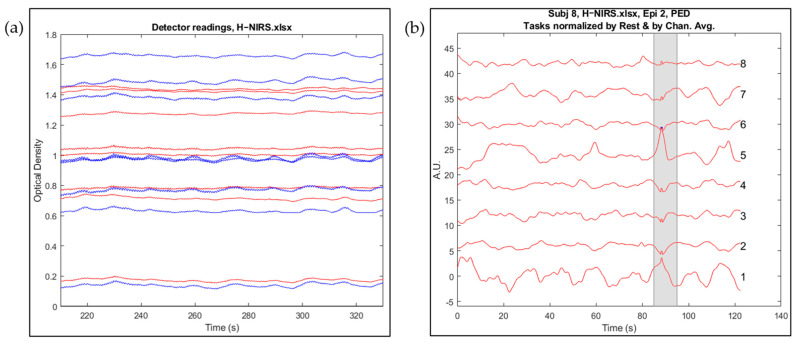
(**a**) Raw fNIRS data; blue and red lines show detector readings at two different wavelengths, i.e., 765 nm and 855 nm, respectively. (**b**) fNIRS data (HbO) after artefact removal and pre-processing are shown for 8 channels. Grey shaded area is identified as an artifact and removed from the data.

**Figure 6 sensors-23-08926-f006:**
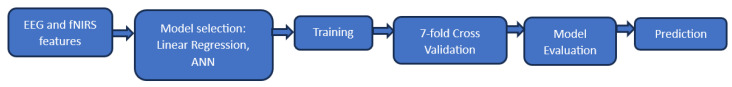
Machine learning using Linear regression and ANN for predicting traditional measures using physiological measures.

**Figure 7 sensors-23-08926-f007:**
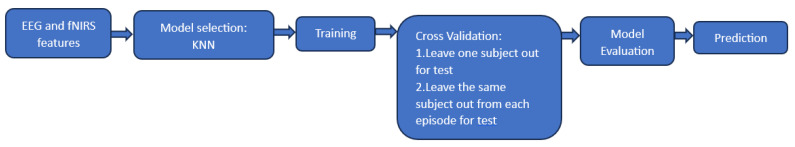
Machine learning using KNN for finding the accuracy of correlation between traditional measures and physiological measures.

**Figure 8 sensors-23-08926-f008:**
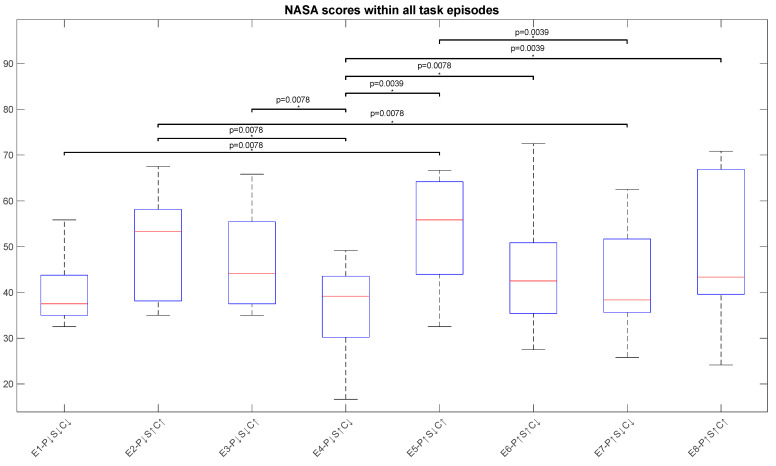
NASA-TLX questionnaire score for all task episodes for all subjects are illustrated. The *p*-values for significant differences, identified by *, are mentioned above the boxes for NASA-TLX scores. The box plot shows the median (red line), the lower and upper quartiles (upper and lower edge of the box), and the minimum and maximum values (black lines extended above and below the box) that are not outliers.

**Figure 9 sensors-23-08926-f009:**
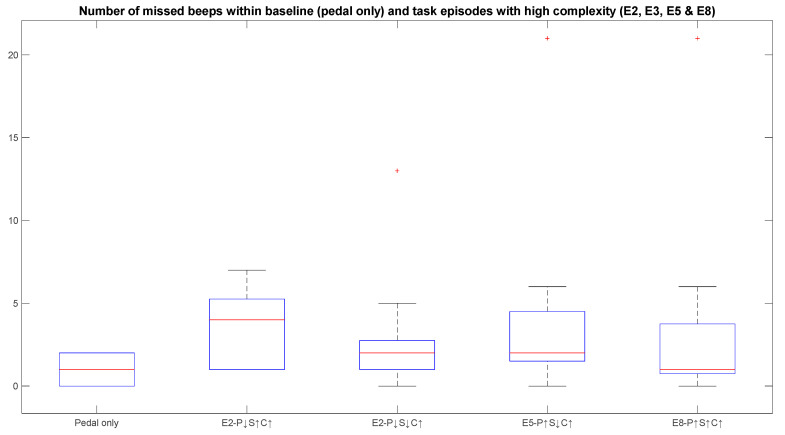
Numbers of missed beeps for all subjects for baseline episode and high-complexity episodes. The box plot shows the median (red line), the lower and upper quartiles (upper and lower edge of the box), outliers (“+” sign) and the minimum and maximum values (black lines extended above and below the box) that are not outliers.

**Figure 10 sensors-23-08926-f010:**
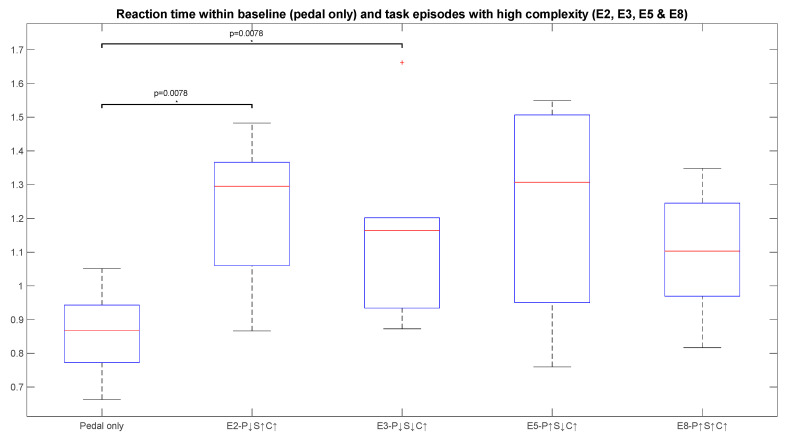
The reaction time of all subjects for baseline episode and high-complexity episodes. The *p*-values for significant differences, identified by *, are mentioned above the boxes for NASA-TLX scores. The box plot shows the median (red line), the lower and upper quartiles (upper and lower edge of the box), outliers (“+” sign) and the minimum and maximum values (black lines extended above and below the box) that are not outliers.

**Figure 11 sensors-23-08926-f011:**
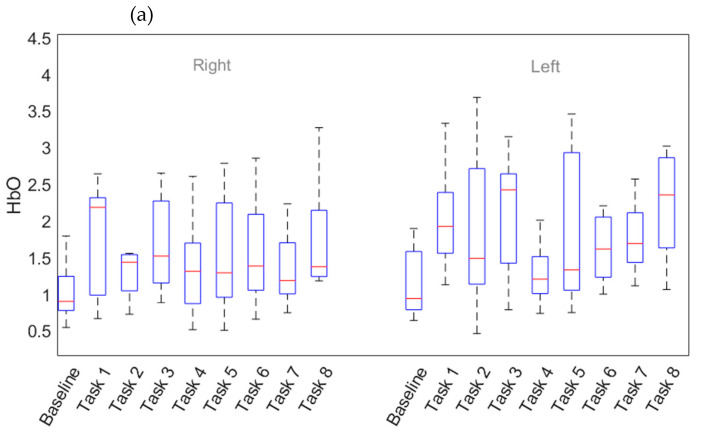
Impact of performance parameters on fNIRS. (**a**) HbO concentration for all tasks in the left and right prefrontal cortex and (**b**) for performance measure. (**c**) HbR concentration for all tasks in the left and right prefrontal cortex and (**d**) for performance measure. The box plot shows the median (red line), the lower and upper quartiles (upper and lower edge of the box), and the minimum and maximum values (black lines extended above and below the box) that are not outliers.

**Figure 12 sensors-23-08926-f012:**
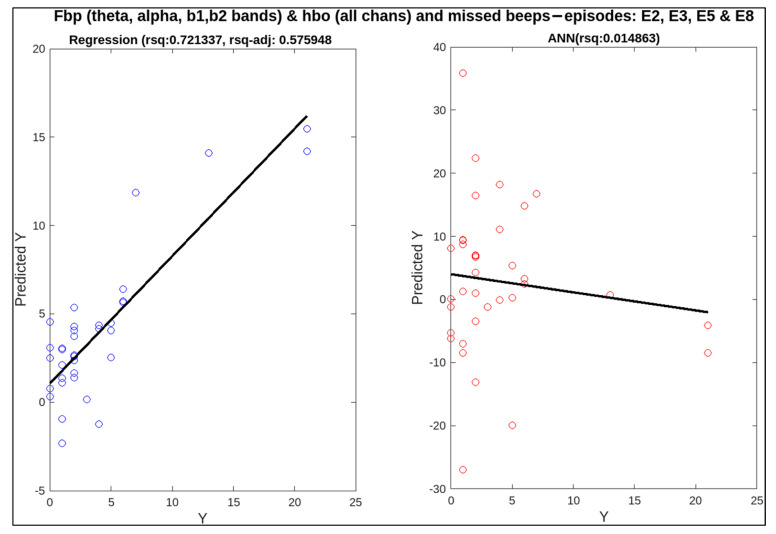
Correlation of neural measures with behavioural measure, i.e., missed beeps, for the highest value of adjusted R-squared (0.654146) for regression.

**Figure 13 sensors-23-08926-f013:**
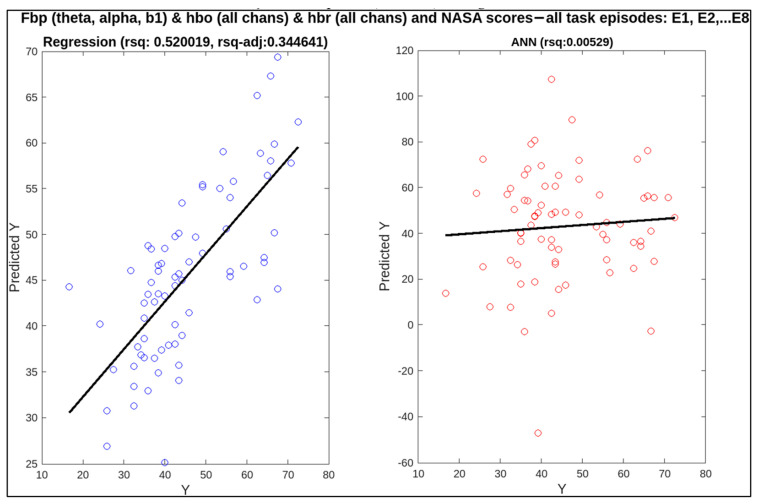
Correlation of neural measures with subjective measure, i.e., NASA-TLX score, for the middle value of adjusted R-squared (0.344641) for regression.

**Figure 14 sensors-23-08926-f014:**
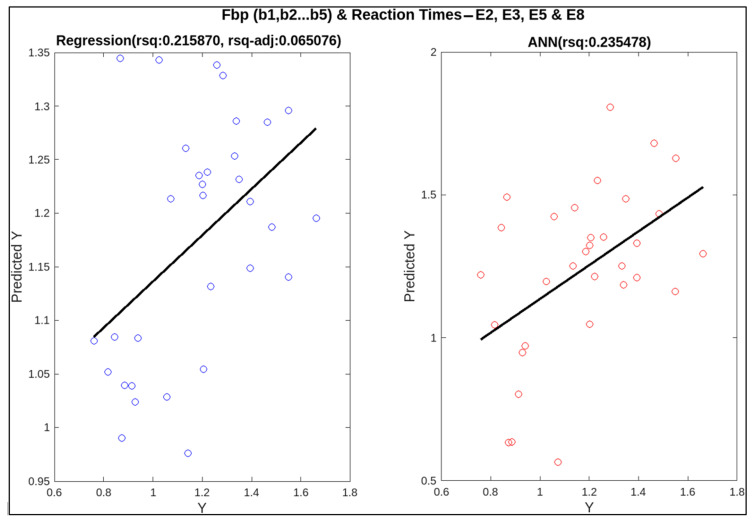
Correlation of neural measures with behavioural measure, i.e., reaction time, for the lowest value of adjusted R-squared (0.065076) for regression.

**Table 1 sensors-23-08926-t001:** Eight different episodes can be seen as a combination of either high or low value for performance measures, i.e., cobot motion speed, task complexity, and payload complexity.

Episode No.	Cobot Speed	Payload Capacity	Task Complexity
1	L	L	L
2	L	H	H
3	H	L	H
4	H	H	L
5	L	L	H
6	L	H	L
7	H	L	L
8	H	H	H

**Table 2 sensors-23-08926-t002:** Individual parameters’ (complexity, cobot speed, and payload capacity) impact on average relative power is observed in this table. Significant *p*-values are highlighted in red ink.

EEG Bands	Payload Changing						Speed Changing						Complexity Changing					
	E1(P↓S↓C↓) vs. E7(P↑S↓C↓)			E2(P↓S↑C↑) vs. E8(P↑S↑C↑)			E1(P↓S↓C↓) vs. E4(P↓S↑C↓)			E5(P↑S↓C↑) vs. E8(P↑S↑C↑)			E1(P↓S↓C↓) vs. E3(P↓S↓C↑)			E6(P↑S↑C↓) vs. E8(P↑S↑C↑)		
	Average Relative Power		*p*-Value	Average Relative Power		*p*-Value	Average Relative Power		*p*-Value	Average Relative Power		*p*-Value	Average Relative Power		*p*-Value	Average Relative Power		*p*-Value
	E1	E7		E2	E8		E1	E4		E5	E8		E1	E3		E6	E8	
delta	0.9113	0.912	0.8203	0.9191	0.9226	1	0.9113	0.9094	0.6523	0.9175	0.9226	0.3594	0.9113	0.9226	0.3594	0.9194	0.9226	0.4258
theta	0.2399	0.2444	0.5703	0.2671	0.2724	0.7344	0.2399	0.2532	0.1641	0.2584	0.2724	0.4258	0.2399	0.2724	0.0273	0.261	0.2724	0.25
alpha	0.1623	0.1654	0.6523	0.1902	0.1979	0.7344	0.1623	0.1621	0.8203	0.1739	0.1979	0.0742	0.1623	0.1979	0.0078	0.178	0.1979	0.0195
beta1	0.1118	0.1056	1	0.1363	0.1478	0.4258	0.1118	0.0982	0.3594	0.1178	0.1478	0.0742	0.1118	0.1478	0.0391	0.1228	0.1478	0.0039
beta2	0.0925	0.0709	0.4258	0.0987	0.1102	0.6523	0.0925	0.0601	0.0977	0.0871	0.1102	0.1641	0.0925	0.1102	0.4258	0.0922	0.1102	0.1289
beta3	0.0744	0.0473	0.25	0.0886	0.0941	1	0.0744	0.0424	0.1289	0.0652	0.0941	0.0977	0.0744	0.0941	0.4258	0.073	0.0941	0.0547
beta4	0.0682	0.0283	0.1641	0.0741	0.0744	0.9102	0.0682	0.033	0.0547	0.0466	0.0744	0.0977	0.0682	0.0744	0.8203	0.0556	0.0744	0.25
gamma	0.0398	0.0077	0.3594	0.0499	0.0492	0.7344	0.0398	−0.0048	0.0391	0.0238	0.0492	0.1289	0.0398	0.0492	1	0.033	0.0492	0.4961

**Table 3 sensors-23-08926-t003:** Pact of the pairwise combination of performance parameters (complexity, cobot speed, and payload capacity) on EEG band power can be seen in this table. Significant *p*-values are highlighted in red ink.

EEG Bands	Payload & Speed Changing						Payload & Complexity Changing						Speed & Complexity Changing					
	E1(P↓S↓C↓) vs. E6(P↑S↑C↓)			E3(P↓S↓C↑) vs. E8(P↑S↑C↑)			E1(P↓S↓C↓) vs. E5(P↑S↓C↑)			E4(P↓S↑C↓) vs. E8(P↑S↑C↑)			E1(P↓S↓C↓) vs. E2(P↓S↑C↑)			E7(P↑S↓C↓) vs. E8(P↑S↑C↑)		
	Average Relative Power		*p*-Value	Average Relative Power		*p*-Value	Average Relative Power		*p*-Value	Average Relative Power		*p*-Value	Average Relative Power		*p*-Value	Average Relative Power		*p*-Value
	E1	E6		E3	E8		E1	E5		E4	E8		E1	E2		E7	E8	
delta	0.9113	0.9194	0.4258	0.9068	0.9226	0.4258	0.9113	0.9175	0.3008	0.9094	0.9226	0.3008	0.9113	0.9191	0.9102	0.912	0.9226	0.1289
theta	0.2399	0.261	0.0391	0.2473	0.2724	0.25	0.2399	0.2584	0.0039	0.2532	0.2724	0.2031	0.2399	0.2671	0.0078	0.2444	0.2724	0.0078
alpha	0.1623	0.178	0.1641	0.1686	0.1979	0.2031	0.1623	0.1739	0.0977	0.1621	0.1979	0.0391	0.1623	0.1902	0.0039	0.1654	0.1979	0.0117
beta1	0.1118	0.1228	0.4961	0.1114	0.1478	0.0742	0.1118	0.1178	0.9102	0.0982	0.1478	0.0273	0.1118	0.1363	0.0039	0.1056	0.1478	0.0078
beta2	0.0925	0.0922	1	0.0772	0.1102	0.25	0.0925	0.0871	0.4258	0.0601	0.1102	0.0547	0.0925	0.0987	0.4258	0.0709	0.1102	0.0391
beta3	0.0744	0.073	0.9102	0.0533	0.0941	0.1641	0.0744	0.0652	0.4258	0.0424	0.0941	0.0547	0.0744	0.0886	0.0742	0.0473	0.0941	0.0117
beta4	0.0682	0.0556	0.5703	0.0392	0.0744	0.25	0.0682	0.0466	0.0977	0.033	0.0744	0.0547	0.0682	0.0741	0.3594	0.0283	0.0744	0.0117
gamma	0.0398	0.033	0.9102	0.0038	0.0492	0.2031	0.0398	0.0238	0.1641	−0.0048	0.0492	0.0977	0.0398	0.0499	0.25	0.0077	0.0492	0.0742

**Table 4 sensors-23-08926-t004:** Combined impact of all performance parameters (complexity, cobot speed, and payload capacity) can be seen in this table. Significant *p*-values are highlighted in red ink.

EEG Bands	Payload, Speed and Complexity Changing		
	E1 (P↓S↓C↓) vs. E8 (P↑S↑C↑)		
	Average Relative Power		*p*-Value
	E1	E8	
delta	0.9113	0.9226	0.3594
theta	0.2399	0.2724	0.0273
alpha	0.1623	0.1979	0.0078
beta1	0.1118	0.1478	0.0391
beta2	0.0925	0.1102	0.4258
beta3	0.0744	0.0941	0.4258
beta4	0.0682	0.0744	0.8203
gamma	0.0398	0.0492	1

**Table 5 sensors-23-08926-t005:** Prediction of conventional measures using brain data variables by employing regression. (~) indicates the highest, medium, and lowest values for R-squared adjusted.

Predictors	Target	Episode	Adjusted R-Squared (Regression)
Theta, alpha, b1	NASA-TLX	All	0.396
FBP (all bands)	NASA-TLX	All	0.4037
HbO	NASA-TLX	All	0.271
HbR	NASA-TLX	All	0.2322
FBP (all bands), HbO	NASA-TLX	All	0.326645
FBP (all bands), HbR	NASA-TLX	All	0.268
Theta, alpha, b1, HbO, HbR	NASA-TLX	All	~0.34464
Theta, alpha, b1	Missed beeps	E2, 3, 5, 8	0.354
FBP (all bands)	Missed beeps	E2, 3,5,8	0.365
HbO	Missed beeps	E2, 3, 5, 8	0.3865
Theta, alpha, b1, b2, HbO	Missed beeps	E2, 3, 5, 8	0.575948
Theta, alpha, b1, b2, HbR	Missed beeps	E2, 3, 5, 8	0.575346
Theta, alpha, b1, HbO, HbR	Missed beeps	E2, 3, 5, 8	0.6057
Theta, alpha, b1, b2, HbO, HbR	Missed beeps	E2, 3, 5, 8	~0.654146
Theta, alpha, b1	Reaction time	E2, 3, 5, 8	0.081942
b1-5	Reaction time	E2, 3, 5, 8	~0.065
HbO	Reaction time	E2, 3, 5, 8	0.262
Theta, alpha, b1, HbO	Reaction time	E2, 3, 5, 8	0.213281
Theta, alpha, b1, HbR	Reaction time	E2, 3, 5, 8	0.262749
Theta, alpha, b1, HbO, HbR	Reaction time	E2, 3, 5, 8	0.259581

**Table 6 sensors-23-08926-t006:** Prediction of subjective and behavioural measures using neural measures with the help of KNN. ** highlights the significant cases.

Target	Physiological Measure (Predictors)	Leave One Subject out for Test %	Statistical Significance of Classification, *p*-Values	Leave the Same Subject out from Each Episode for Test %
NASA-TLX	All Fbps, HbO, HbR	56.9	0.0860	52.8
	All Fbps	69.4	0.0007 **	54.2
	All Fbps, HbO	63.9	0.0072 **	56.9
	All Fbps, HbR	54.2	0.1840	50
	HbO	66.7	0.0016 **	58.3
	HbR	52.8	0.2385	50
Reaction Times	All Fbps, HbO, HbR	62.5	0.0355 **	59.4
	All Fbps	65.6	0.0368 **	37.5
	All Fbps, HbO	53.1	0.2025	59.4
	All Fbps, HbR	62.5	0.0344 **	65.6
	HbO	56.3	0.1275	43.8
	HbR	62.5	0.0354 **	65.6
Missed Beeps	All Fbps, HbO, HbR	72.2	0.0088 **	69.4
	All Fbps	75	0.0062 **	66.7
	All Fbps, HbO	75	0.0034 **	72.2
	All Fbps, HbR	77.8	0.0012 **	72.2
	HbO	75	0.0031 **	72.2
	HbR	77.8	0.0012 **	72.2

## Data Availability

Data supporting reported results can be found published in this paper.

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
