# Peer review of "Multimodal Assessment of Cognitive Workload Using Neural, Subjective and Behavioural Measures in Smart Factory Settings"

_sensors, 2023, doi:10.3390/s23218926_

Round 1

Reviewer 1 Report

Comments and Suggestions for Authors

This paper is surely well written and the methodology is clear and sound. I consider it ready for publication. However, I would just suggest the Authors to check the difference between cognitive workload and mental workload in literature and ISO standards before finalising this document. 

Author Response

Thank you for your valuable comments and suggestions. Please find attached the document containing point by point response to your comments.

Reviewer 2 Report

Comments and Suggestions for Authors

This study proposed machine learning models to predict the factory workers mental workload using physiological, behavioural, and subjective measures. A total of 9 participants were participated in this study. Their classification accuracy up to 77.8%.

The following comments are provided for the authors’ reference.

1.     [Abstract] Please stated the classification performances in the abstract

2.     [Page 3] Please provide the experimental scene of using cobot

3.     [Page 3-4] The participant characteristic should be mentioned in detail

4.     [Table 1] What are the parameter of the eight episodes examined in this study.

5.     [Page 5, line 201] Provide the significance of reaction time, missed beeps, and Stroop task error rate.

6.     [page 5, line 223] Please check the down-sampled of the EEG signals. Is it from 200Hz to 200Hz?

7.     [Page 6, line 256] The normalization of the hemoglobin concentration is divided by standard deviation or subtracted with mean and divided by standard deviation?

8.     [Page 6, line 267] Why the authors did not consider all physiological, subjective, and behaviour as predictors? The target could be specified by “Low” or “High” by expert judgment.

Author Response

Thank you for your valuable and insightful comments and suggestions. The manuscript has been updated accordingly. Please find attached the document containing point by point response to your comments.

Reviewer 3 Report

Comments and Suggestions for Authors

In this manuscript, the authors analyzed different factors that affect human-robot interactions regarding human factors. EEG and fNIR were recorded and compared with other measures to predict worker's workload. Linear regression and ANN were used to determine the correlation between physiological data and behavioural measures. 
The manuscript can be improved by adding technical illustrations of the experimental setup and the classification models.

1. Major comments: as a submission to this journal, I would add figures to clearly demonstrate how measurement was taken, and then how the machine learning model was built.

2. Line 178 and Figure 1, what is the difference between episode 1 and episode 2,3...? Episode 2,3 seems to have secondary tasks implemented in parallel? Why?

2. Line 193, is there evidence to show that wearing EEG and fNIR instruments does not affect the stress, fatigues...?

3. Line 224, from 200 Hz to 200 Hz, typo?

4. Section 2.3, for this work, and also as a submission to this journal, I think it is crucial to see how good the artifact removal method is. Can the authors provide an extra figure showing how EEG, fNIR was processed and also a comparison of raw and processed data? An optional suggestion to this is to add the experiment setup (how EEG and fNIR instruments were equiped,  where the co-robot locates?)

5. Line 330, as the author indicated, the participant becomes familiar with the cobot along the progress of the experiment. How to factor in this condition when analyzing performance variables? Especially, does the analysis in section 3.1 hold for worker-cobot interactions in the long-term?

6. I might miss it but did you explain p value in Figure 2 somewhere when you introduce the episode difference?  Maybe add it to the caption too?

7. Suggestion: why not make Table 2 a multi-line graph showing EEG FSP changes alongside different episodes?

8. In table 3, regression model's highest R2 is around 0.65, is the model well trained for the task? Or maybe I misinterpretate though. This question apply for ANN and KNN as well. I see improvements by combining physiological parameters, but are the models generally good enough to support the claim for classification?

Author Response

(The authors gave the same response as above.)

Reviewer 4 Report

Comments and Suggestions for Authors

First and foremost, I would want to express my admiration for the comprehensive and profound nature of your study report. The investigation into the collaboration between humans and cobots is both urgent and crucial, and your research offers useful insights into this rapidly developing domain.

The author's rigorous approach to the experimental setup and the thorough analysis provides a comprehensive perspective on the assessment of cognitive workload in manufacturing environments. I commend the diligence and meticulous approach you have employed.

With that being stated, I am of the opinion that there are specific elements that, if attended to, can further refine and augment the influence of your research. Please see below the key aspects that I would appreciate your consideration of:

Details of the participants: I would appreciate further elucidation of the individuals involved in the study. Could you kindly furnish a comprehensive demographic profile, encompassing pertinent background information, including their experience within the manufacturing area, as well as any other important factors that may potentially impact the cognitive workload?

Sample Size Justification: The explanation for selecting a sample size of 9 participants for the research is of interest, as it is important to comprehend the reasoning behind deeming this sample size as enough for the analysis. In order to strengthen this decision, it would be beneficial to include statistical arguments or references from pertinent academic literature.

A two-minute period of rest. Duration: The duration of the 2-minute rest time between each experimental phase captured my attention. Could you provide some clarification regarding the underlying reasoning for selecting this particular duration? I am interested in comprehending the impact of rest length on following outcomes and whether any measures have been taken to account for cognitive load reset or normalization during these periods.

The examination of limitations is an essential component of any research endeavor, since it acknowledges the inherent constraints that are present in any study. Acknowledging these factors not only enhances the paper's credibility but also provides valuable guidance for future studies. It would be advantageous to incorporate a section that explicitly addresses the limitations or obstacles encountered throughout this research endeavor.

Author Response

(The authors gave the same response as above.)

Round 2

Reviewer 2 Report

Comments and Suggestions for Authors

The authors have well-addressed the reviewer comments and improved the manuscript.

Author Response

Thank you for your thoughtful and valuable comments which helped in improving the manuscript. 

Reviewer 3 Report

Comments and Suggestions for Authors

Thanks for addressing my comments.

Please check the following:

Figure 3, for EEG, is down sampling after the band pass filtering?

Figure 4, top left annotation, "boundary" and "5sec" overlapped. Please seperate them.

Author Response

Thank you for your thoughtful and insightful comments and suggestions. Please find attached the document containing point-by-point response of all comments. 
